



# Cloud mask algorithm from the EarthCARE multi-spectral imager: the M-CM products

Anja Hünerbein[1], Sebastian Bley[1], Stefan Horn[1,3], Hartwig Deneke[1], and Andi Walther[2]

[1]Leibniz Institute for Tropospheric Research, Leipzig, Germany
[2]Cooperative Institute for Meteorological Satellite Studies, Madison, WI, United States
[3]Meteologix AG, Sattel, Switzerland

**Correspondence:** Anja Hünerbein (anjah@tropos.de)

**Abstract.**

The EarthCARE satellite mission will provide new insights into aerosol-cloud and radiation interactions by means of synergistic observations of the Earth's atmosphere from a collection of active and passive remote sensing instruments, flying on a single satellite platform. The Multi-Spectral Imager (MSI) will provide visible and infrared images in the cross-track direction with a 150 km swath and a pixel sampling at 500 m. The suite of MSI cloud algorithms will deliver cloud macro- and microphysical properties complementary to the vertical profiles measured from the ATmospheric LIDar (ATLID) and the Cloud Profiling Radar (CPR) instruments. The MSI cloud mask algorithm (M-CM) provides a cloud flag, cloud phase and cloud type product which are essential parameters for the cloud optical and physical properties (M-COP) as well as for the aerosol optical properties (M-AOT). Synthetic test scenes from the EarthCARE end-to-end simulator provide a first overview of the expected performance of the M-CM products before launch. Validation of the MSI cloud mask algorithm against satellite products from MSG SEVIRI and MODIS demonstrates a good performance of the algorithm. The MSI cloud detection performance is not completely aligned with that from MODIS, which is however not surprising when considering the rather limited number of spectral channels of MSI in comparison to MODIS. The direct comparison to MODIS shows that some threshold tuning should be done especially over desert or sun glint regions once real observations are available. However the MSI bands are less affected by sun glint because of the asymmetric view of the imager.

## 1 Introduction

Clouds cover about 70 % of our Earth's surface and play an important role in the global radiation and energy budgets. The influence of clouds on radiative fluxes exhibits a complex dependency on cloud type, phase, geometric height as well as their optical and microphysical properties, potentially introducing significant radiative feedbacks in response to climate change. The IPCC Sixth Assessment Report summarizes the current state of knowledge, concluding that clouds are expected to amplify global warming as a result of an increase in high-level clouds, and a reduction in low-level clouds (IPCC 2021, in press). The report provides a best estimate of the net cloud feedback, having a positive value of 0.42 W m$^{-2}$. While the uncertainty related to cloud feedbacks has been halved compared to the previous fifth assessment report, the response of clouds to a warming Earth remains one of the biggest challenges in our understanding of the climate system.





The determination of cloud, atmospheric and surface properties from multi-spectral satellite imagery relies on the accurate discrimination of cloudy and cloud-free pixels. This discrimination is typically done by a cloud mask algorithm as the first step in a processing chain of satellite imagery. Small errors in the cloud mask can lead to large uncertainties, misrepresentations and misinterpretations in subsequently derived products. Many different cloud detection techniques have been developed in the past. These techniques are mostly based on two general assumptions, namely that clouds appear brighter in solar channels

due to the strong reflection of sunlight, and colder in infrared channels relative to cloud-free surfaces, due to the decrease of atmospheric temperature with height. In addition, discrimination of clouds from cloud-free regions is commonly based on a variety of spectral features, spatial structure measures, or temporal characteristics in time series, because clouds are often more variable than the underlying surface (Saunders and Kriebel, 1988).

Operational cloud mask algorithms generally combine a variety of individual tests by means of a a decision tree, as no single

test is able to achieve a sufficient accuracy for the diversity of clouds and atmospheric conditions encountered globally (e.g. Saunders and Kriebel, 1988). An alternative is the use of fuzzy-logic based or Bayesian schemes to combine tests to yield a confidence value or probability for the classification (e.g. Ackerman et al., 1998; Hollstein et al., 2015). More recently, convolutional neural networks have been applied to discriminate between different land surfaces, ocean, clouds and cloud shadows (Mateo-García et al., 2017; Li et al., 2019; Hughes and Kennedy, 2019). Such cloud masking approaches are often applied to

high-resolution satellite images (e.g, Landsat, Sentinel-2), and require large training datasets. In practice, these training datasets have to be created manually, and the significant effort required for establishing high-quality training datasets and validating their performance has so far not led to operational application in global-scale long-term cloud climate data records. Rossow and Garder (1993) classify the different tests used in cloud mask algorithms into radiance threshold tests, spatial variance tests, temporal variance tests and tests using independent datasets to estimate clear-sky radiances. The performance of these tests

strongly depends on the satellite sensor specifications including spatial, spectral and temporal resolution.

The International Satellite Cloud Climatology Project (ISCCP, Schiffer and Rossow, 1983) was the earliest effort to provide a comprehensive global cloud climatology from multi-spectral meteorological satellite imagers. Its cloud detection algorithm is described in Rossow and Garder (1993), and is based on a combination of static and dynamic threshold tests for one window channel in the visible and one window channel in the thermal infrared wavelength range. This choice was made based on the

limited availability of channels from early geostationary satellites, specifically the Meteosat, GMS (Geostationary Meteorological Satellite) and GOES (Geostationary Operational Environmental Satellite) series.

Based on the Advanced Very High Resolution Radiometer (AVHRR) flown on NOAA's polar-orbiting satellites since the early 1980's, the APOLLO (AVHRR processing scheme over clouds, land, and ocean) cloud detection scheme used both static and dynamic threshold tests. The availability of additional spectral channels was used in particular to improve night-time cloud

detection performance. Dynamic thresholds were derived from a histogram-based scene analysis (Saunders and Kriebel, 1988; Strabala et al., 1994).

A new milestone in instrumental capabilities was reached by the MODerate-resolution Imaging Spectrometer (MODIS) instrument, providing observations in 36 spectral channels from NASA's Earth Observing System satellites Terra and Aqua launched in 1999 and 2002, respectively. The operational cloud mask product for MODIS considers the spectral information



from 19 of these channels (Ackerman et al., 2002; Platnick et al., 2003). While several spectral tests are similar to those used by the APOLLO and ISCCP cloud detection schemes, the availability of channels in water vapor and CO2 absorption bands enabled an improved cloud detection in particular for thin high-level clouds and for polar night conditions (e.g. Liu et al., 2004; Nakajima et al., 2011).

EarthCARE, the Earth Clouds Aerosols and Radiation Explorer, is a joint European and Japanese mission and part of ESA's
Living Planet Program (Illingworth et al., 2015; Wehr et al., 2022). The mission objective is to improve our understanding of cloud-aerosol-radiation interactions, and the role of aerosols and clouds in the Earth radiation budget. While observation of clouds have gradually improved over the past decades, the launch of the EarthCARE satellite is expected to bring a breakthrough by means of its novel observational capabilities. To achieve the mission objective, accurate and simultaneous measurements of microphysical and optical properties of aerosol and clouds together with solar and infrared radiation fluxes
are crucial. EarthCARE will offer the unique opportunity to collect these observations at global scale due to its polar orbit. The satellite will carry an exceptional collection of active and passive remote sensing instruments, flying on a single satellite platform in an orbit at an altitude of 393 km. The instruments include the ATmospheric LIDar (ATLID), the Cloud Profiling Radar (CPR), the Multi- Spectral Imager (MSI) and the BroadBand Radiometer (BBR).

This manuscript describes the algorithm used to produce the cloud flag, type and phase products based alone on MSI
observations. The approaches selected for EarthCARE's MSI cloud mask (M-CM) products relies on the research on and experience with cloud masking approaches during the past 40 years since the start of the satellite era. It exploits the full spectral information content of the MSI instrument, and also includes a histogram-based scene analysis. It is however important to realize that its performance is also determined by the selection of 4 solar and 3 infrared channels for MSI, having central wavelengths of 670 nm (VIS), 865 nm (NIR), 1650 nm (SWIR-1), 2210 nm (SWIR-2), 8.8 μm (TIR-1), 10.8 μm (TIR-2), and
12.0 μm (TIR-3). Given this specification, MSI's capabilities and sensitivity is more similar to that of AVHRR than of MODIS. In particular, no channels within absorption bands of atmospheric gases are available. Reflectances in the solar channels are used to detect clouds by means of a visible reflectance test and a reflectance ratio test. The visible reflectance test assumes that the reflectance of clouds exceeds the reflectance of cloud-free surfaces, with the exception of highly reflective surfaces. The reflectance ratio test compares the ratio of the reflectances of two shortwave channels to thresholds. Complementing the solar
channel tests, a brightness temperature test uses information from the thermal infrared (TIR) channels to detect clouds based on the assumption that the brightness temperature of clouds is significantly lower than the brightness temperature of cloud-free pixels.

The estimation of the expected difference in cloud-free brightness temperatures for the three infrared channels is an important aspect for the accuracy of cloud detection. This difference depend on differences in atmospheric absorption (water vapor) and
surface emissivity. Therefore, scene-dependent look-up tables or online radiative transfer simulations have to be elaborated to determine suitable thresholds. All tests yield a probability that a pixel is cloud-free. Some of the individual tests are however not independent from each other, because they rely on similar channels and principles. Hence, the resulting probabilities of those tests are combined.



For every 500 m resolution pixel of the 150 km-wide MSI swath, the M-CM products provide a classification whether it is cloud-covered or cloud-free as final output. Additionally, for the cloudy pixels, the cloud type and cloud phase of the uppermost cloud layer will be reported.

This paper is structured as follows. Section 2 describes the algorithms for deriving the operational Level 2 M-CM products, which comprise a binary cloud flag, cloud phase, cloud type and confidence statistics. The verification of the algorithm using MODIS and MSG SEVIRI scenes as well as synthetic test data from the EarthCARE End-to-End simulator (Donovan et al., 2022) is provided in section 3. Comprehensive comparisons between the operational M-CM product and the synthetic test fields are presented in the appendix. The data processing chain including the role of M-CM is explained in more detail in Eisinger et al. (2022).

## 2 M-CM algorithm description

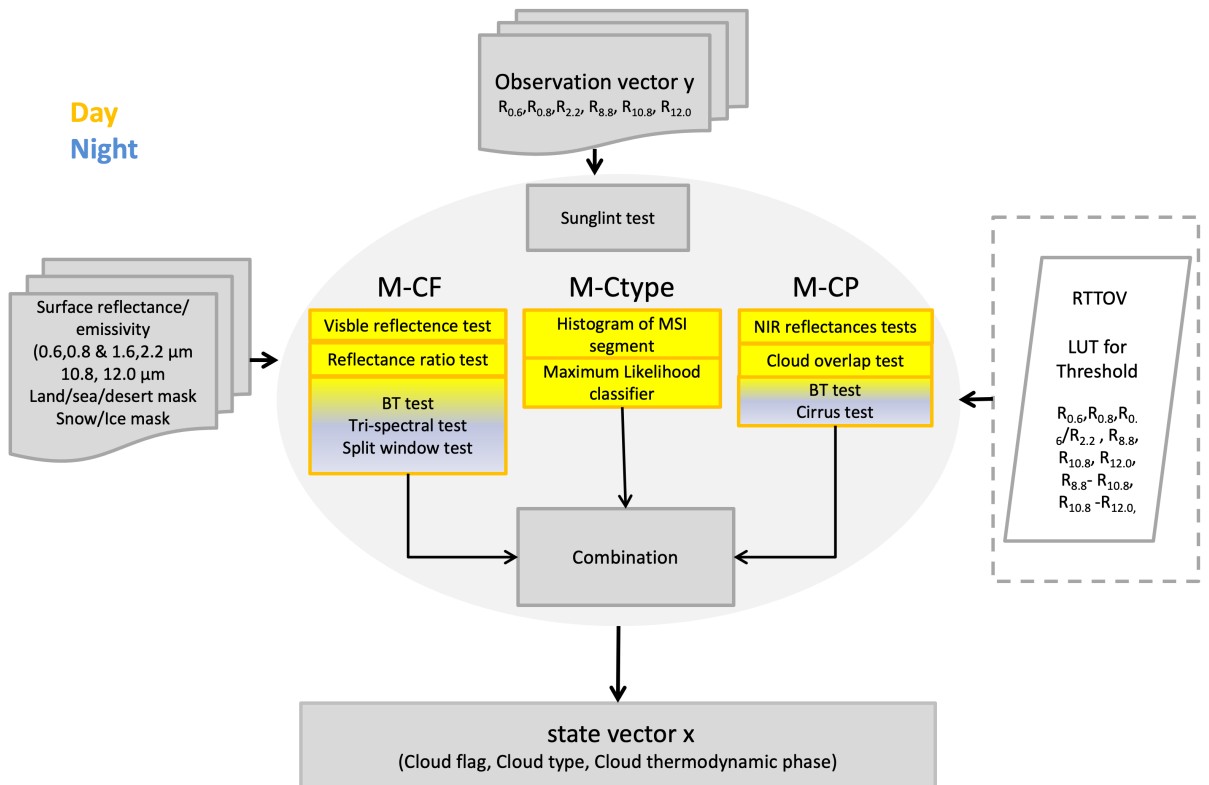

**Figure 1.** Schematic of the main components of the M-CM algorithm.

The MSI cloud product processor (M-CLD) provides algorithms for calculation of the cloud flag, cloud phase, cloud type, cloud optical depth, cloud particle size, cloud water path and cloud top temperature/pressure/height. The processor consists





of two main parts, which are sequentially processed. First is the cloud mask (M-CM) which is mandatory for the other cloud optical and physical properties (M-COP). The present manuscript describes the cloud mask processor (M-CM), which is schematically shown in Fig. 1.

The algorithm starts with the calculation of the reflectances at the top of the atmosphere in the shortwave channels. The reflectances ($\rho_i$) of each channel $i$ are obtained from the measured radiance ($L$) and the solar irradiance $E_0$ as

$$\rho_i(\Theta_0, \Theta, \varphi) = \frac{\pi L_i(\Theta_0, \Theta, \varphi)}{E_0 cos(\Theta_0)}, i = 0.6, 0.8, 1.6, 2.2 \tag{1}$$

with the sun zenith angle $\Theta_0$, the viewing zenith angle $\Theta$ and the relative azimuth angle $\phi$.

An important input for the algorithm is the day/night flag. Day time condition is considered for a certain pixel of the sun zenith angle $\Theta_0 < 80°$. Additionally, the sun glint angle $\Theta_r$ is calculated over ocean as

$$cos(\Theta_r) = sin(\Theta) * sin(\Theta_0) * cos(\Phi) + cos(\Theta) * cos(\Theta_0) \tag{2}$$

If $\Theta_r < 36°$, the pixel is flagged as sun glint provided in the surface flag.

## 2.1 M-CF: Binary cloud flag

The algorithm derives a cloud mask by applying individual threshold tests to brightness temperatures and reflectances of individual channels. The threshold tests and the way how results are combined are adopted from the MODIS cloud mask algorithm (Ackerman et al., 2002). The thresholds rely on the assumption that spectral signatures of cloud-free pixels and pixels covered by different cloud types differ. Consequently, each threshold test is dedicated to a certain cloud type. Because it makes no sense to apply a strict classification where a pixel is assumed to be cloudy if the test score exceeds a certain threshold and to classify it as cloud-free if the value is lower than the threshold, there is a probability for each pixel of being cloud-free derived.

The probability of being cloud-free from the applied tests is combined to an overall probability which may provide in combination with the number of applied tests a measure of the confidence of the result. From the overall probability a binary cloud mask indicating if a pixel is cloudy or not is derived.

### 2.1.1 Visible reflection tests

The visible reflectance test compares the reflectance in the $0.67\,\mu m$ channel or the reflectance in the $0.865\,\mu m$ channel with surface dependent thresholds (Fig. 2). If the reflectance exceeds the upper threshold pixels are assumed to be very likely cloudy. Pixels with reflectances below the lower threshold are with high confidence cloud-free. The pixels in-between are classified by calculating probability functions, as described in section 2.1.3.

The upper and the lower thresholds differ for land, desert, ocean pixels outside the sun glint region and ocean pixels in the sun glint region (Fig. 2). Whereas the thresholds are fixed for the first three classes, they depend on the sun glint angle in the sun glint region. Over land the test applies the reflectance in the $0.67\,\mu m$-channel, while over desert the reflectance in the $0.865\,\mu m$-channel is used. Ocean pixels located outside the sun glint region are classified by using the reflectance in the




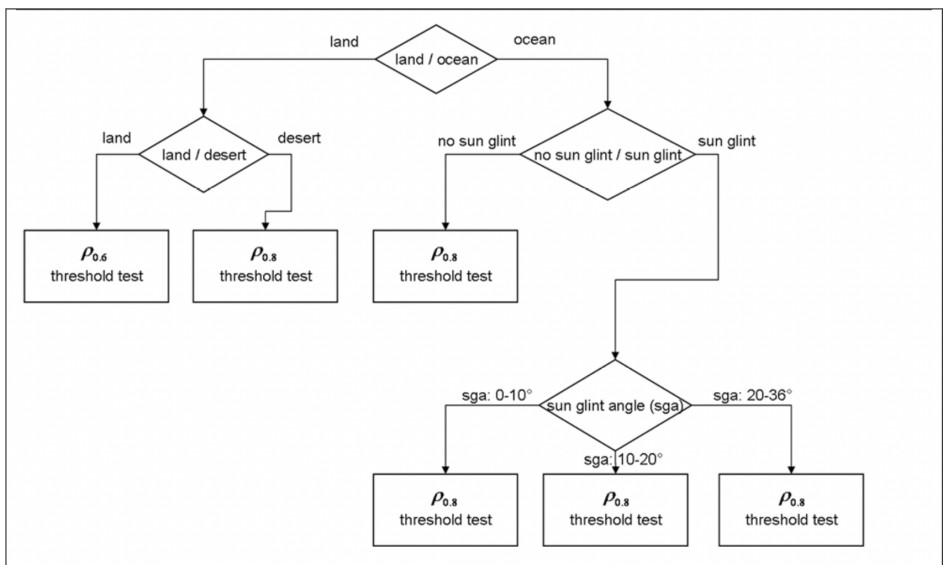

**Figure 2.** Flow chart of visible reflectance test.

0.865 $\mu$m-channel. Ocean pixels affected by sun glint also apply thresholds based on the 0.865 $\mu$m-channel, but the thresholds are calculated dependent on the sun glint angle. The lower and upper thresholds of the 0.865 $\mu$m-tests depend on predefined limits of sun glint angles between 0°–10°, 10°–20° and 20°–36° (Fig. 2).

### 140 2.1.2 Reflectance ratio test

The reflectance ratio test is applied to daytime pixels over oceans and land surfaces with low reflectivities. Therefore, the land pixels are classified in surfaces with high reflectivity like desert, polar and semi-arid region and low reflectivity.

Over ocean the reflectance ratio test can be applied as well in the sun glint region. The test score is the ratio of the reflectance in the 0.865 $\mu$m-channel and the reflectance in the 0.67 $\mu$m-channel. If the test score is smaller than the lower threshold the

pixel is high confidentially cloud-free. A test score larger than the upper threshold labels the pixel as high confidential cloudy. For pixels with values in-between, the confidence level is calculated in a linear way. Upper and lower thresholds are defined for ocean pixels outside and inside the sun glint region, respectively.

For land pixel indicated by the application mask as appropriate, the test score is a modified GEMI (Global Environmental Monitoring Index) first described by Pinty and Verstraete (1992). It is calculated as

$$m\_gemi = \eta(1 - 0.25 \cdot \eta) - \frac{\rho_{0.6} \cdot 100 - 0.125}{1 - \rho_{0.6} \cdot 100} \tag{3}$$

with

$$\eta = \frac{2(\rho_{0.8} \cdot 100 - \rho_{0.6} \cdot 100) + 1.5 \cdot \rho_{0.8} \cdot 100 + 0.5 \cdot \rho_{0.6} \cdot 100}{\rho_{0.8} \cdot 100 + \rho_{0.6} \cdot 100 + 0.5} \tag{4}$$





If $m\_gemi$ is greater than $m\_gemi_{clear}$, the pixel is high confidentially clear and if $m\_gemi$ is lower than $m\_gemi_{cloudy}$, the pixel is assumed to be high confidentially cloudy. If values in-between appear, then confidence level of being clear is calculated
by a linear approach.

### 2.1.3   Brightness temperature tests

We use two different approaches for the brightness temperature tests, one using simple thresholds and the other one applying brightness temperature differences between different infrared channels for the separation between cloudy and cloud free pixels.

The first simple threshold test is applied on the $10.85\,\mu$m-channel for all surface types during night-time.

The pixels identified as cloudy if

$$T_{10.8} < T_{10.8\_cs} \tag{5}$$

where the clear sky brightness temperature, $T_{10.8\_cs}$ at top of the atmosphere, is calculated with the IR radiative transfer model (RTTOV, (Saunders et al., 1999)) on the grid of the auxiliary meteorological (X-MET) data and then interpolated to the geolocation and measurement time of the MSI pixel. The X-MET dataset provides additional meteorological model parameters
required for the processing (Eisinger et al., 2022).

The details about the RTTOV forward simulation is described in Hünerbein et al. (2022). If $T_{10.8_{cs}}$ is larger than $T_{10.8}$, the pixel is assumed to be cloudy.

The probability of being cloud-free is calculated by assuming a linear probability function.

The tri-spectral window brightness temperature difference test (at $8.8\,\mu$m, at $10.8\,\mu$m and at $12.0\,\mu$m) is only applied to
water surfaces during day time. The brightness temperatures at $10.8\,\mu$m and at $12.0\,\mu$m are used to detect thin cirrus clouds and cloud edges, which are characterized by a higher brightness temperature difference ($10.8\,\mu$m - $12.0\,\mu$m) than cloud free surface. The pixel is detected as cloudy if:

$$T_{10.8} - T_{12.0} > T_{diff1\_cs} \tag{6}$$

where $T_{diff1\_cs}$ is calculated with RTTOV for each pixel for clear sky conditions.
By use of the temperature differences on $8.8\,\mu$m -$10.8\,\mu$m thin cirrus clouds over all surface conditions can be detected. In addition to equation 6 if the difference is relatively high compared to the clear sky condition then the pixel is classify as cloudy if:

$$T_{8.8} - T_{10.8} > T_{diff2\_cs} \tag{7}$$

The probability of being cloud-free is calculated by assuming a linear probability function. The same applies for the tri-spectral
brightness temperature difference test that further investigation is needed to define the base threshold, which are strongly dependent on surface and water vapor.





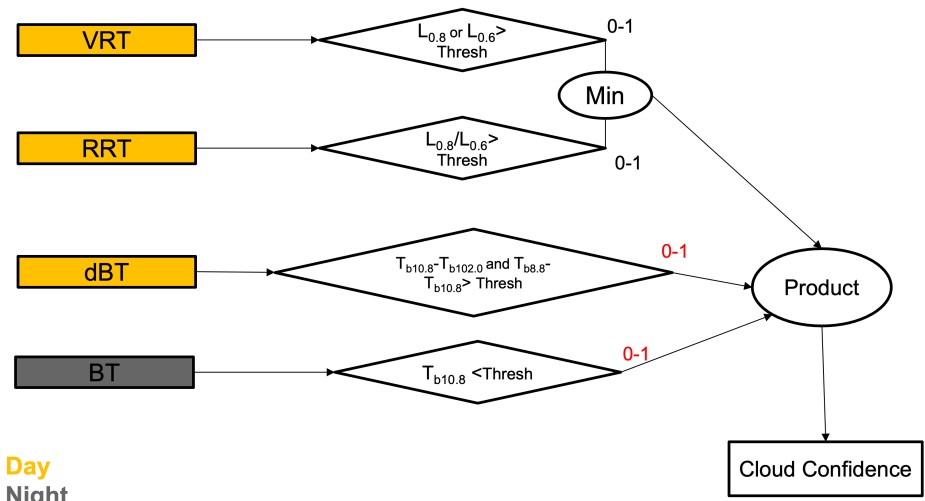

**Figure 3.** Four groups of cloud tests to make cloudy confidences.

### 2.1.4 Estimation of confidence level

The results of all tests are combined in a two-step procedure for determination of the confidence level (Fig. 3). In the first step the overall probability for each pixel from the tests applying reflectances is derived because these tests are not independent. This is accomplished by finding the minimum probability $G_i$ of being cloudy in both tests. In the next step the probability from the brightness temperature test and the intermediate result from the reflectance tests are combined by calculation of the square root of the multiplied values if multiple valid test results are available:

$$Q = \sqrt[n]{\prod_{i=1}^{N} G_i} \tag{8}$$

Otherwise the final result consists of the valid test result or is undefined. The square root of the multiplied probabilities that the pixel is clear ensures that the overall result does not tend to cloudy pixels as it would be the case if results where solely multiplied. This approach is considered clear-sky conservative.

### 2.2 M-Ctype: Cloud types

The algorithm applies a Maximum-Likelihood-Classifier to reflectances and brightness temperatures at VIS, SWIR-2 and TIR-2.

Before the algorithm assigns a specific cloud type for a certain pixel, the dataset needs to be trained to acquire statistics for predefined cloud classes. This procedure is described in the following section.



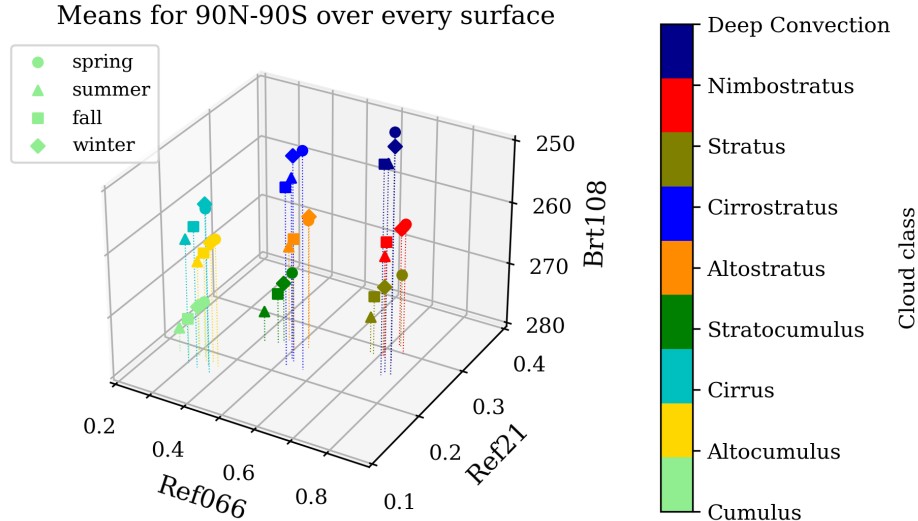

**Figure 4.** Observed reflectances and brightness temperatures at VIS, SWIR-2 and TIR-2 (MODIS) for the 9 ISCCP cloud classes (cirrus, cirrostratus, deep convection, altocumulus, altostratus, nimbostratus, cumulus, stratocumulus and stratus) and seasonal separation.

### 2.2.1 Cloud type training using MODIS

A large number of MODIS scenes are used to learn statistics for nine predefined cloud classes (from thin to thick clouds, high, medium and low clouds) and one cloud free class, either over sea, land or desert covers and separated into stripes of 15 degree

latitude. Nine cloud classes are categories by using the MODIS cloud top height and cloud optical thickness based on the ISCCP cloud classification schemes (Rossow and Schiffer, 1999). From these scenes, the mean vector and covariance matrix are calculated for all cloud classes and one cloud free class from the visible channel, the shortwave-infrared channel and the infrared channel and saved in a look up table.

The region, season and surface are identified for each pixel. The regions are defined by circle of latitude in 15° steps.

The pixels are separated in four seasons (winter, spring, summer and autumn) based on the month (Fig. 4). The surfaces are separated with the land sea mask in land, water and desert pixel. The nine ISCCP cloud classes can be clearly distinguished between cirrus, cirrostratus, deep convection, altocumulus, altostratus, nimbostratus, cumulus, stratocumulus and stratus. Also a clear sky class is defined for the different surface types, regions and seasons (not shown in Fig. 4).

The statistics are then used to assign each pixel in the measured scene to a certain class by applying a Maximum Likelihood

Classifier. The algorithm assumes either a completely cloud covered or completely cloud-free pixel and does not take sub-pixel clouds into account.



### 2.2.2 Maximum Likelihood classifier

The probability is computed for each MSI pixel to all individual classes by means of a Maximum Likelihood classifier. A pixel is assigned to class $j$ if the likelihood of class $j$ is the greatest among the $9 + 1$ classes which are relevant for the respective surface. The maximum likelihood is found by

$$j = argmax \left[ f(x|m_i \sum_i) \right] \qquad (9)$$

$$f(x|m_i, \sum_i) = \frac{1}{\sqrt{(2\pi)^p \left| \sum_i \right|}} exp \left[ \frac{-1}{2} (x - m_i)^T \sum_i^{-1} (x - m_i) \right] \qquad (10)$$

with $x$ the vector of properties (reflectances and brightness temperature) in the considered channels, $m_i$ the mean vector of class $i$ and $\sum_i$ the covariance matrix and $p$ the number of Maximum Likelihood classes for the respective surface.

Though also Maximum-Likelihood-Classifier exist that does not assign a class when the maximum probability value falls below a certain probability, the classifier applied here is a hard classifier assigning a class to every pixel with valid radiation data independent of the magnitude of the maximum probability.

The reliability of a maximum likelihood classification result depends on the probability $p_i = f(x|m_i, \sum_i)$ for the assigned class $i$ and the probability for the next class $j$ derived as $p_j = f(x|m_i, \sum_j)$. The next class is determined by minimizing $argmin = [|p_j - p_i|] = j$.

The assignments to the nine cloud classes and the clear sky class are determined for all pixels.

### 2.3 M-CP: Cloud phase

The discrimination of the thermodynamic phase at cloud top is based on the spectral absorption differences in ice and water clouds between the visible (0.67 $\mu$m) and the shortwave infrared (1.65 $\mu$m) as well as the brightness temperatures at 8.8, 10.8 and 12.0 $\mu$m. The cloud phase categories of the M-CP algorithm include liquid water, ice, supercooled mixed-phase and cloud overlap (e.g. multi-layer clouds). The M-CP retrieval closely follows the approach applied to AVHRR and VIIRS (Pavolonis et al., 2005; Pavolonis and Heidinger, 2004) as well as for MODIS (Strabala et al., 1994).

The algorithm consists of several spectral threshold tests applied to the reflectances from the VIS, SWIR and TIR channels. The thresholds are adapted from the corresponding AVHRR channels based on Pavolonis et al. (2005). The fine tuning of these thresholds will be done with the whole measurements suite of EarthCARE at nadir.

The algorithm starts with a series of threshold tests based on TIR-2, which follows the physical assumption the the cloud top phase depends on the cloud top temperature. The liquid water category includes clouds of liquid water droplets that have a temperature greater than 273.16 K measured by TIR-2. Only non opaque cirrus clouds can also fall in that category. To detect semitransparent cirrus clouds over optically thick water clouds, a cloud overlap test is done. The cloud overlap detection uses the VIS, TIR-2 and TIR-3 channels. This method is adapted from the AVHRR algorithm explained by Pavolonis and Heidinger (2004). The underlying physical theory is that the VIS reflectance will not change much when having an overlapping thin





cirrus cloud over a thick water cloud, while the temperature difference between both clouds results in a brightness temperature difference of the IR window channels that is larger than predicted by radiative transfer calculations. A certain pixel is defined

as ice cloud, if the BT at 10.8 $\mu$m < 233.16 K and the overlap test fails. Supercooled mixed-phase cloud pixels are assumed based on threshold tests with the BT at 10.8 $\mu$m between 233.16 K and 273.16 K.

During daytime conditions, additional tests are applied using the SWIR-1 channel, which improves the detection of overlapping and cirrus clouds.

## 2.4 Surface flag

The surface flag distinguishes between water, land, desert, vegetation, snow, sea ice, sun glint and undefined. While the surface types water, land, desert and sun glint are used as input for the M-CM algorithm, the types vegetation and snow are calculated for the cloud free pixels only, by using the Normalized Differenced Vegetation Index (NDVI) and the Normalized Different Snow Index (NDSI).

The NDVI is the normalized ratio of the difference in reflectance at NIR and VIS based on the red edge feature of the

vegetation. The NDSI is the normalized ratio of the difference in reflectance at VIS and SWIR-1. The atmosphere is transparent at both wavelengths, while snow is very reflective at VIS and not reflective at SWIR-1.

The algorithm distinguishes between sparse vegetation/ocean and dense vegetation with the NDVI and identifies snow surfaces with the NDSI.

## 2.5 M-CM Quality flags

The M-CM quality flags provide pixel-based quality information for the cloud flag, the cloud type and the cloud phase products. The quality flags distinguish between high, medium, low and poor quality. These measures do not represent probabilities, but rather the number of tests which have been executed for the associated pixel, the consistency among the products or the surface flag. The definitions of the individual quality flags are provided in table 1.

| Quality status | Cloud flag | Cloud type | Cloud phase |
|---|---|---|---|
| **High** | All tests executed and results consistent with M-Ctype | Results consistent with M-CF | Results consistent with BT thresholds for water (BT<233.16 K) and ice (BT>273.16 K) |
| **Medium** | All tests executed and results inconsistent with M-Ctype | Surface flag is ocean | Surface flag is ocean |
| **Low** | Less than 50 % of the tests were executed | Surface flag is land | Surface flag is desert |
| **Poor** | Only one test executed (e.g. for night) | Surface flag is desert | Only night tests performed |

**Table 1.** Definitions of pixel-based quality flags (high, medium, low, poor) for the cloud flag, the cloud type and the cloud phase products.



The results of the M-CF and M-Ctype are also combined to a final cloud mask quality flag. A high-quality flag means that
both results are consistent.

## 3 Verification of the M-CM algorithm performance

The algorithm performance and processing chaining has been tested by applying the M-CM processor to scenes from the
MODIS and MSG SEVIRI instruments and to synthetic atmospheric test scenes created with the EarthCARE End-to-End
simulator (Donovan et al., 2022).

### 3.1 Validation against synthetic test scenes

Three specific synthetic test scenes have been created based on forecasts from the Global Environmental Multiscale Model (Qu,
2022) to test the full chain of EarthCARE processors (Donovan et al., 2022). These test scenes cover a variety of atmospheric
situations over ocean, land and ice surface during day and night time. The natural colour RGB images of the three test scenes
are provided in the appendix A.

It seems very appealing to validate our cloud algorithm against the test scenes, however the results should be handled with
care because they strongly depend on the assumptions made in the model. But since no observational EarthCARE-like dataset
exists, the synthetic model dataset provides the best available proxy for testing the EarthCARE processing chain and synergistic
products (Donovan et al., 2022).

The most prominent one is the HALIFAX scene covering a 6000 km long frame starting over Greenland, crossing the eastern
end of Canada and ending in the Caribbean (Fig. 5). The scene starts over the Greenland ice sheet with mixed-phase clouds
at night-time, transitioning from deeper clouds with tops up to 6 km around $65°$N to mixed-phase clouds with tops around
3 km at temperatures as cold as $-30°$C over the eastern edge of Canada. Below there is a high ice cloud regime followed by a
low-level cumulus cloud regime embedded in a marine aerosol layer below an elevated dirty dust layer around 5 km altitude.
The original model outputs are generated for 7 December 2015 using the Canadian GEM model (Qu, 2022).

While the binary cloud flag and cloud phase product provide results for the high-latitude part of the Halifax scene, the cloud
type product does not show results there. This is due to the night-time conditions. The Maximum-Likelihood classifier requires
also information in the visible bands, which makes it impossible to classify cloud types during night-time. For the cloud flag,
only brightness temperature tests have been applied. For this reason, the cloud mask quality flag indicates only poor quality
there.

### 3.1.1 Validation of the M-CF cloud flag with 3D model output

The M-CF cloud flag is validated against the output from the 3D model fields (Donovan et al., 2022). The model cloud flag is
calculated based on the extinction profiles at 680 nm from the model output, which we consider as the truth. In the first step,
we have calculated the cloud optical thickness (COT) as the extinction of radiation along the path from the earth surface to the



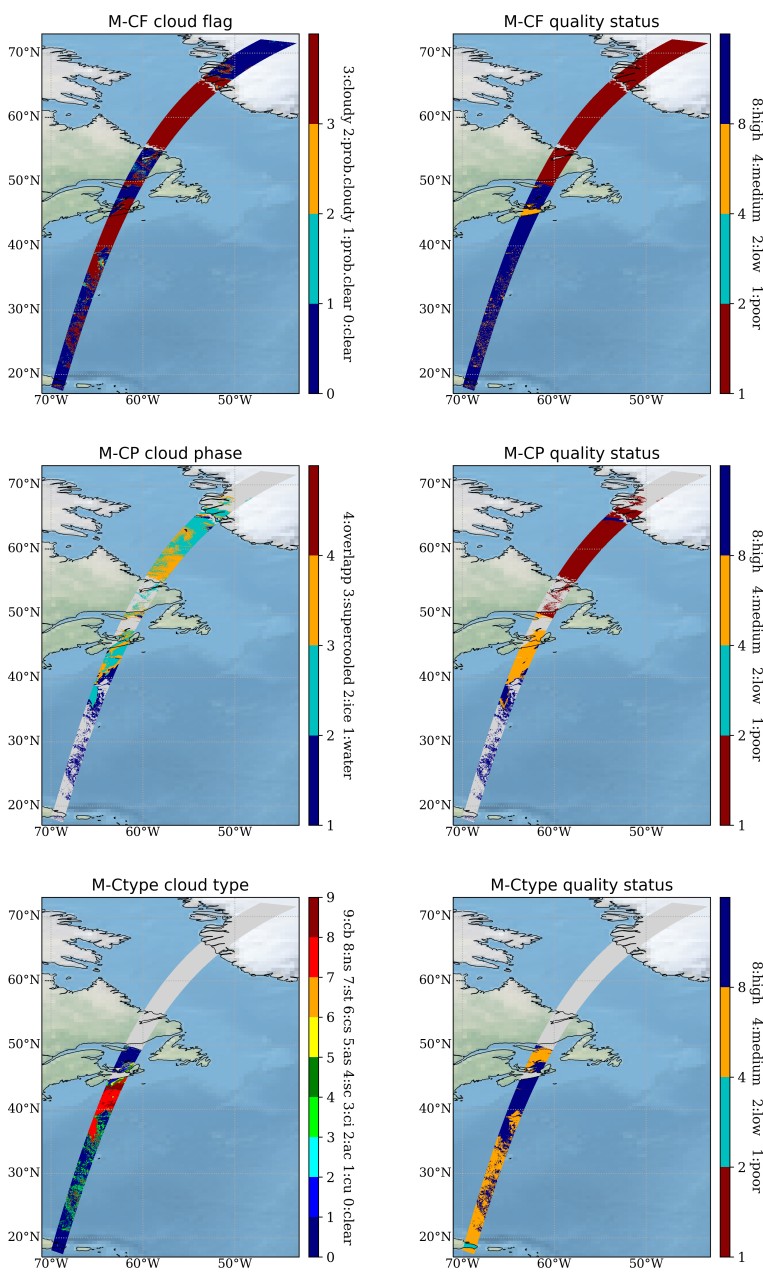

**Figure 5.** M-CM processor applied to the Halifax scene including the binary cloud flag (M-CF) and cloud mask quality flag (upper row), the cloud phase (M-CP) and quality flag (middle row) and cloud types (M-Ctype) and quality flag (bottom row). The light grey shaded region indicates pixels, labeled as undefined by the processor.

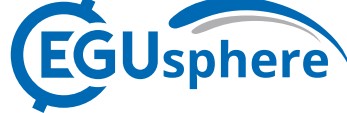

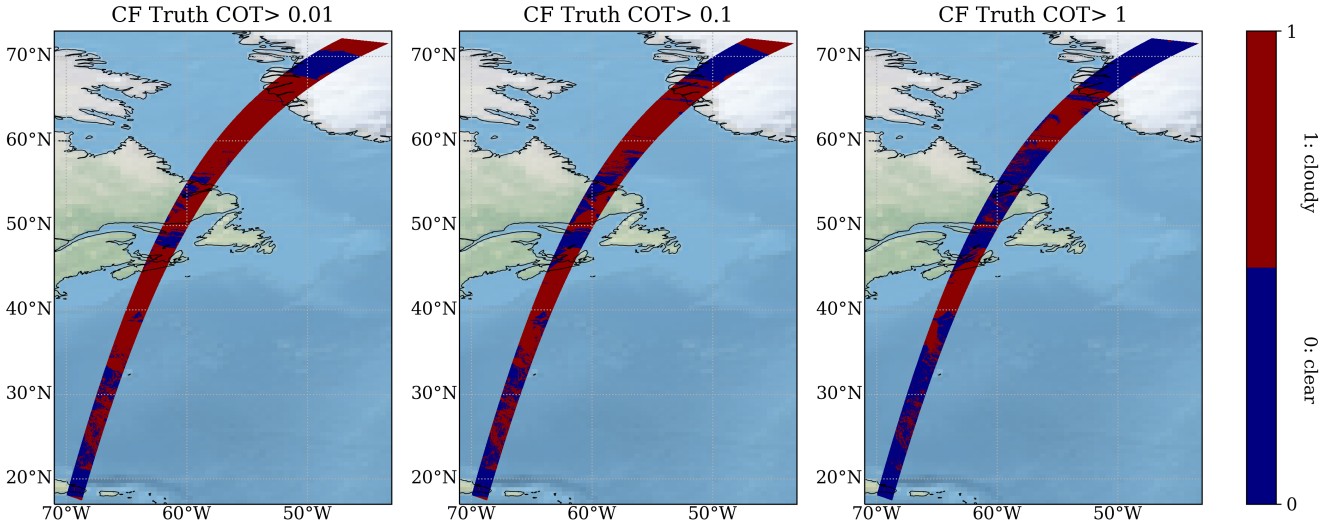

**Figure 6.** True cloud flag based on the 3D extinction fields at 680 nm for the Halifax scene. Cloud-free and cloudy areas are identified by applying three different thresholds on the column integrated cloud optical thickness, COT≥0.01 (left), COT≥0.1 (middle) and COT≥1 (right). The resulting cloud fraction is 72 % (COT≥0.01), 61 % (COT≥0.1) and 37 %(COT≥1).

top of atmosphere at 680 nm. The second step defines a certain profile as cloudy applying three different threshold, COT≥0.01,
COT≥0.1 and COT≥1.

Figure 6 shows the true cloud flag, based on the 3D extinction profiles for three different thresholds applied to the COT. When assuming pixels with COT≥0.01 to be cloudy, the overall cloud fraction of the scene would be 72 %. The cloud fraction decreases to 61 % for COT≥0.1 and to 37 % for COT≥1. This demonstrates that the true cloud mask is very sensitive to the choice of the COT threshold. The M-CF algorithm yields a cloud fraction of 50 %. The best agreement between the cloud
fraction of the true cloud flag and the M-CF cloud flag is achieved when applying a threshold of COT≥0.1. The cloud detection sensitivity of the M-CF algorithm is clearly better than COT≥1, but in contrast to COT≥0.1, a few cloudy pixels with probably optically thin clouds are not detected by the M-CF cloud flag.

Figure 7 illustrates the differences between the M-CF cloud flag and the true cloud flag based on the 3D model output considering a threshold of COT≥0.1, which is assumed to be the best choice. Both cloud flags are in good agreement for most
parts of the scene. Only a few false cloudy pixels are visible over the ocean which are most likely thin clouds with COT≤0.1 that are detected by MSI, but not in the cloud flag for COT≥0.1. The orange pixels in the centre of the scene show pixels that are detected as clear-sky by M-CF, while the true cloud flag defines them as cloudy. This can be explained by the fact that different thresholds are applied for snow and land surface types, but there are inconsistencies between the surface types in the M-CF algorithm and the model data. The M-CF algorithm uses surface information from the X-MET data as input, while the
model data uses slightly different surface specifications. The scattered false clear pixels in the lower part of the scene are due to edge pixels of low level clouds, which are not detected by the M-CF cloud flag.





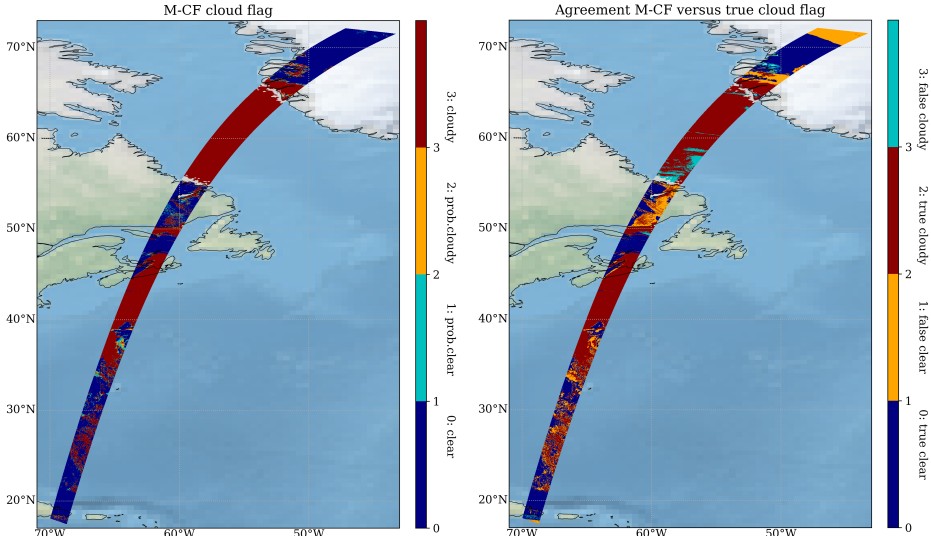

**Figure 7.** M-CF cloud flag (left) and differences between the binary M-CF and the true cloud flag for the Halifax scene (right). The M-CF cloud fraction is 50 %, while the true cloud flag results in a cloud fraction of 61 %.

## 3.2 Validation against MODIS

| Algorihm | M-CF | M-Ctype | M-CF + M-Ctype | MODIS |
|---|---|---|---|---|
| Cloud fraction (%) | 52 | 41 | 69 | 80 |

**Table 2.** Comparison of the scene cloud fraction between M-CF, M-Ctype, the combination of M-CF and M-Ctype and MODIS.

The M-CM cloud mask algorithm has also validated against MODIS scenes. In contrast to the synthetic scenes, the MODIS scenes do not rely on the assumptions made in the background model. We have used MODIS Terra L1b calibrated radiances (MOD021KM) as well as global forecast data form the Copernicus Atmosphere Monitoring Service (CAMS) as input for the M-CLD processor. Although MSI and MODIS share similar channel characteristics, differences in the cloud products are expected due to differences in the central wavelength and spectral response functions.

Figure 8 shows the MSI M-CM cloud flag and the MODIS cloud flag for an example over West Africa on 11 September 2021 at 11:50 UTC. Both cloud flags discriminate between clear-sky, cloudy, probably cloudy and probably clear. The false color RGB image uses the MODIS bands 1 (620-670 nm), band 4 (545-565 nm) and band 3 (459-479 nm). The MSI surface flag separates between water (1), land (2), desert (3), vegetation (4), snow (5,6), sea ice (7), sun glint (8) and undefined (0), the scene over West Africa has no snow or sea ice pixels. Both types, desert and sun glint represent difficulties for cloud masking algorithms, which is why the largest differences between the MODIS and MSI cloud flag are found over these surface types. The MSI cloud flag yields a cloud fraction of 52 %, while MODIS results in 80 %. When converting the M-Ctype cloud classes





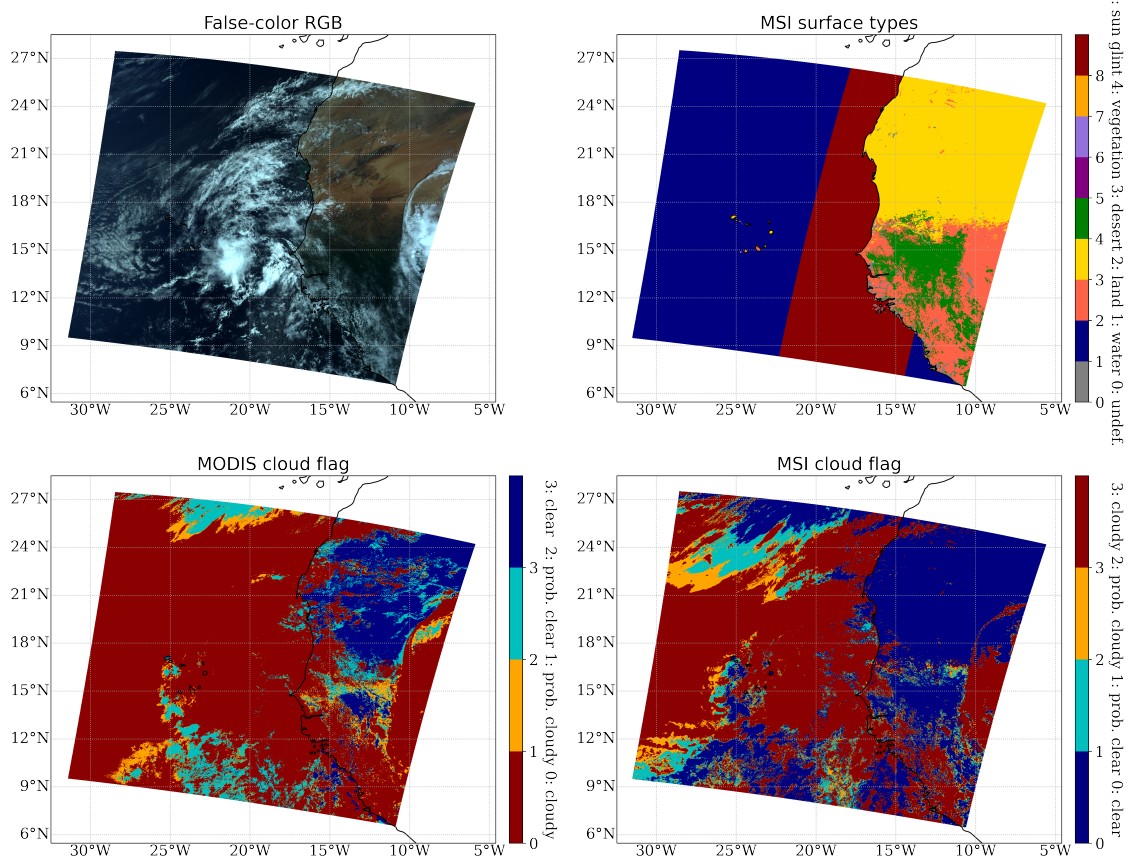

**Figure 8.** M-CM algorithm applied to satellite data from MODIS over West Africa on 11 September 2021 at 11:50 UTC. The MODIS false color RGB composite (using MODIS bands 1,4 and 3) is shown on the top left, the MSI surface flag on the top right, the MODIS cloud flag on the lower left and the MSI cloud flag on the lower right. The MSI surface types 5, 6 and 7 are snow and sea ice flags, which are not present in the present case study. While 1 (water), 2 (land), 3 (desert) and 8 (sun glint) are inputs for the processor, type 4 (vegetation) is based on the NDVI and only calculated for clear-sky pixels in the M-CF flag.

in a binary cloud class results in a cloud fraction of 41 %. The product is independent from M-CF, because it uses a Maximum Likelihood classifier. When combining both binary M-CM cloud flags into one, the cloud fraction increases to 69 % (Tab.2). This result demonstrates that the combination of both independent M-CM cloud products leads to a better agreement with MODIS than just using one of them. The MSI algorithm misses large parts of clouds over desert, but also over the ocean in the





upper part of the scenes there are clear differences. These differences are expected because the MODIS cloud tests are based
on much more spectral channels. For the majority of clouds, which are visible on the RGB image, the agreement between the
MODIS and MSI cloud flag is very good.

To get more robust statistics, the cloud mask comparison has been done for the full month of September 2021. The MSI
cloud flag systematically shows a lower cloud fraction than the MODIS cloud flag. Only in cases with a strong sun glint effect,
the combined M-CF and M-Ctype cloud mask shows a higher cloud fraction than MODIS.

For assessing the overall agreement between the MSI and MODIS cloud mask, we have calculated the percentage of consistency for both clear sky and cloudy for all 45 MODIS scenes in September 2021. The results are shown in table 3.

| Algorithm | M-CF vs. M-Ctype | M-CF vs. MODIS | M-Ctype vs. MODIS | M-CF + M-Ctyp vs. MODIS |
|---|---|---|---|---|
| full scene | 80 % | 76 % | 65 % | 79 % |
| no sunglint | 92 % | 90 % | 87 % | 91 % |

**Table 3.** Comparison of the two M-CM cloud flags and the MODIS cloud flag. The agreement in % is calculated for a binary cloud flag only, where confident cloudy and probably cloudy is merged to cloudy and for clear sky the same. For M-Ctype, all cloud types 1–9 are considered as cloudy.

We have intercompared the M-CF versus M-Ctype products, M-CF versus MODIS, M-Ctype versus MODIS and M-CF
combined with M-Ctype versus MODIS. The overall agreement between M-CF and MODIS is 76 %. This results increases
to 79 % when combining M-CF and M-Ctype (Tab. 3). When excluding all pixels that are labelled as sun glint by the M-
CM surface flag, the agreement increases to 91 %. This finding demonstrates that large parts of the discrepancies are due to
differences in the handling of the algorithms in scenes effected by sun glint, which will be further investigated by tuning the
thresholds with real measurements.

### 3.3 Validation against MSG SEVIRI

The M-CM cloud mask was also part of a cloud retrieval intercomparison study in the framework of the International Cloud
Working Group (ICWG). The ICWG supports the assessment of cloud retrievals applied to passive imagers onboard of geostationary satellites (Hamann et al., 2014; Wu et al., 2017). Therefore, the M-CM algorithm has been applied to images from the
SEVIRI instrument onboard Meteosat Second Generation (MSG). As in the comparison with MODIS, also SEVIRI provides
similar channel characteristics like the MSI imager. Nevertheless, one should be aware that this leads to uncertainties through
the adaptation to SEVIRI due to differences in central wavelength and spectral response function, radiative transfer simulations
and generated look-up-tables.

Twelve groups provided cloud mask data for the SEVIRI disk. The individual input data has been transformed into a binary
cloud mask separating between cloudy and cloud-free. The M-CM cloud mask was with a cloud fraction of 52% in the range of
the other results ranging from 31% to 64%. Figure 9 shows the discrepancies of the different cloud masks results. A pixel value
of 0 means, that all algorithms are in agreement that it is cloudy. Grey values indicate that all algorithms consistently label the



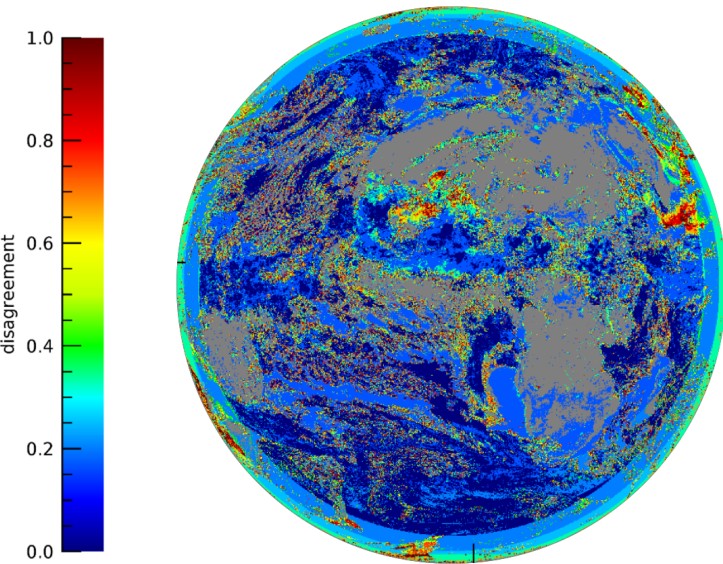

**Figure 9.** Differences between the cloud masks of 12 algorithms for the MSG SEVIRI disk on 13 April 2008. A value of 0 means, that all algorithms for this particular pixel set it as cloudy. The grey values mean that all algorithm agree set this pixel as cloud free. In total the disagreement measure is normalized to 1, if the half of the algorithm classifies a pixel as cloudy and the other half as cloud free.

pixel as clear sky. A disagreement value of 1 shows that half of the algorithms classified a pixel as cloudy and the other half as cloud free. The main discrepancies between the different cloud masks are found to be over Northern Africa, caused by different detection thresholds for thin cirrus clouds over bright surface like desert in this case. It could also be biomass burning aerosol that is classified as clouds by some algorithms. Another area of disagreements is the southern part of the Arabian Peninsula and the adjacent sea.

**4   Conclusions**

In this paper, the algorithms used by the cloud mask processor (M-CM) for the MSI imager onboard EarthCARE are described. The algorithms provide the cloud flag (M-CF), cloud phase (M-CP) and cloud type (M-Ctype) products. The cloud flag and cloud phase at the cloud top are based on spectral threshold tests for the visible and infrared channels, while the cloud type product is based on a maximum likelihood classifier. While the cloud type product is only available during day-time, the cloud

flag and cloud phase products are also retrieved during night-time, although with a reduced number of tests (Fig. 1). The M-CM products are an important input for the subsequent retrieval of the cloud optical and physical products (M-COP) (Hünerbein et al., 2022) and the aerosol optical properties (M-AOT) (Docter et al., 2022).



In order to test the algorithm performance and to get a better impression of the products before the EarthCARE launch, the M-CM algorithm has been verified in this study against synthetic test scenes from the EarthCARE end-to-end simulator and
satellite data from MODIS and MSG SEVIRI.

Using synthetic test data, it is found that the M-CM products are in good agreement with the products from other processors within the EarthCARE instrument suite and with the 3D model fields used as input to the simulator which can thus be considered as the truth. One should keep in mind that in contrast to ATLID or CPR which provide vertical profile information, MSI is a passive instrument that retrieves cloud properties at cloud top or for aerosol and optically thin clouds, total column integrated
information. The synergistic products using data from MSI and ATLID will help to better understand some of the uncertainties of the MSI products (Haarig et al., 2022).

An overall agreement of 79 % was found between the MSI and the MODIS cloud flag using one month MODIS data over Cape Verde in September 2021. The agreement significantly improves to 91 % when excluding the sun glint areas from the comparison. Ocean areas characterized by sun glint represent some of the most challenging scenes for cloud masking
algorithms. This indicates that further adjustments are needed for the thresholds of the M-CM cloud flag for sun glint conditions to improve the performance. However the MSI images are less affected by sun glint in comparison to MODIS due to the fact that the MSI imager is tilted sideways, with 35 km of the full 150 km swath on the sun-facing side and 115 km on the other side of the nadir track.

The M-CM algorithm has also been applied to measurements from MSG SEVIRI, and the results have been compared
against other cloud mask algorithms in the frame of the international cloud working group. The comparison has demonstrated that the M-CM performance lies in the range of the other cloud masks.

Planned improvements of M-CM will include dynamic thresholds for the threshold tests for the cloud flag. We propose that this tuning should be done post-launch once real observations will be available. A tuning of the M-CM thresholds towards better agreement with MODIS is not optimal in the current state, because of the spectral differences between MSI and MODIS. While
MSI features 7 spectral bands, MODIS has 36 spectral bands allowing a better cloud detection performance. The advantage of the MSI observations are in contrast to MODIS, that MSI is flying together with active instruments (e.g. ATLID and CPR) on the same platform which will allow unique synergies of cloud products from different instruments.

The algorithm verification in the present study uses synthetic test scenes and data from other satellite platforms as basis. During the validation phase after the EarthCARE launch, dedicated campaigns will be conducted using ground-based and
airborne instruments, which will offer the opportunity for a more comprehensive validation of the MSI cloud products. Also geostationary satellites will be used for the validation to support the selection of suitable validation datasets and to provide complementary reference datasets on a global scale. Meteosat Third Generation (MTG) will be launched in 2022 in the geostationary orbit (Holmlund et al., 2021) offering with its flexible combined imager (FCI) with 16 spectral channels and up to 500 m spatial sampling excellent opportunities for the validation and synergies with the MSI products.
Further improvements of the M-CM product are expected once real observations are available due to its flexible design based on configuration files, which allows easy adjustment e.g. of cloud mask thresholds without modifying the source code of the whole algorithm chain.

In contrast to the pre-launch MSI test data presented in this study, the MSI spectral bands are affected by a shift of the central wavelength depending on the instrument viewing angle. This effect is caused due to imperfections in the bandpass filters on the curved optical lenses (Wehr et al., 2022; Wang et al., 2022, e.g.). Investigations are ongoing to mitigate this effect in the Level 2 M-CLD and M-AOT retrievals. During the validation phase, aircraft measurements with high-spectral resolution will further help quantifying the impact of the central wavelength shift on the MSI cloud and aerosol products.

*Data availability.* The EarthCARE Level-2 demonstration products from simulated scenes, including the MSI cloud mask products discussed in this paper, are available from https://doi.org/10.5281/zenodo.7117115 (van Zadelhoff et al., 2022).

## Appendix A: Natural colour RGB images of the synthetic test scenes

For the three test scenes a natural colour RGB images are generated to visualise several types of atmospheric and surface features. The natural colour RGB is composed of the VIS, NIR and the SWIR-1 channel data. The images have been linearly stretched within the reflectance ranges to the full range of display values from 0-255 bytes to improve the contrast.

The benefit is the easy interpretation because most of the colours of the image are very similar to a true colour image of the Earth. Figure A1 shows the RGB images for the three test scenes, which includes clouds, snow, vegetation, sun glint and clear skies. Snow on the ground as well as ice over mountains, frozen lakes and sea ice appear cyan in the RGB images (Fig. A1, b). The more homogeneous the snow/ice cover is, the brighter the cyan colour will be. Snow and ice on mountains will therefore be depicted in a stronger cyan colour than snowy surfaces on ground which are often disrupted by vegetation. In addition, clouds with ice crystals appear as well cyan in the RGB images (Figure A1, a) as the ice crystals reflected at $0.67$-$\mu$m and $0.865$-$\mu$m and absorb solar radiation at $1.65$-$\mu$m. Further the different cloud heights, ice crystal habits and sun zenith angles lead to inhomogeneous colour pattern. The ocean and lakes in the RGB images appear in dark-black colour (Fig. A1, c). Vegetation is indicated by green colours because of the stronger reflection of solar radiation at $0.865$-$\mu$m than at $0.67$-$\mu$m (Fig. A1, a, e.g. Halifax Caribbean island). For detailed information how to interpret RGB images see the RGB-color-guide (Eumetrain, 2022).

*Author contributions.* The manuscript was prepared by AH, SB and HD. The M-CM code was developed by AH and SH. AW generated the dataset and created the plots for the ICWG results.

*Competing interests.* The authors declare that they have no conflict of interest.

*Acknowledgements.* This work has been funded by ESA grants of (IRMA), 4000112018/14/NL/CT (APRIL) and 4000134661/21/NL/AD (CARDINAL).



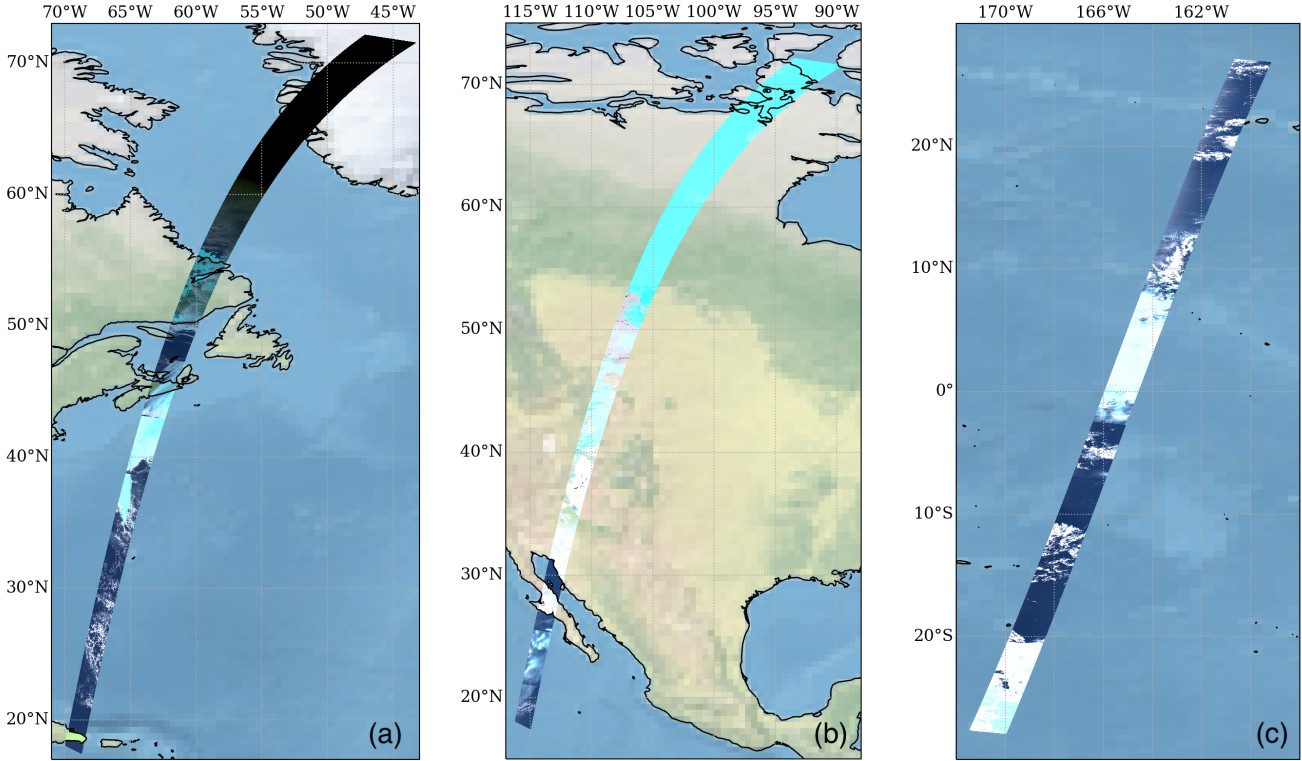

**Figure A1.** Natural colour RGB images generated from the MSI VIS, NIR and SWIR-1 channels for (a) HALIFAX, (b) BAJA and (c) HAWAII (Donovan et al., 2022).

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
