# Peer review of "Cloud mask algorithm from the EarthCARE multi-spectral imager: the M-CM products"

_EGUsphere, 2022_

## Author Comment (AC1)

**Reply to Anonymous Referee #1**

We would like to thank Anonymous Referee #1 for their helpful comments. Below are the original comments in normal with our responses in *italic* text.

**Comments to the author**:
In the study "Cloud mask algorithm from the EarthCARE multi-spectral imager: the M-CM products" by Hünerbein et al. the MSI cloud mask algorithm is described and its performance compared to selected test scenes of the EarthCARE simulator. A first validation shows a good agreement of the M-CM algorithm with 45 MODIS scenes as well as 12 MSG SEVIRI cloud mask algorithms of one selected scene. A description and first tests of the M-CM algorithm is very valuable for the cloud community and on time before the launch of EarthCARE that will provide simultaneous measurements of clouds and aerosol properties as well as radiation fluxes. The paper is written in a clear and concise way. Some minor deficits are noted in the abstract and conveyed messages of the paper. It is recommended for publication after minor revisions.

Minor comments:

The abstract needs modification as the study's purpose is not clearly stated. What is the study's motivation, goal, result and outlook?

*Thanks, the abstract has been adjusted to:*

*"The EarthCARE satellite mission will provide new insights into aerosol-cloud and radiation interactions by means of synergistic observations of the Earth's atmosphere from a collection of active and passive remote sensing instruments, flying on a single satellite platform. The Multi-Spectral Imager (MSI) will provide visible and infrared images in the cross-track direction with a 150 km swath and a pixel sampling at 500 m. The suite of MSI cloud algorithms will deliver cloud macro- and microphysical properties complementary to the vertical profiles measured from the ATmospheric LIDar (ATLID) and the Cloud Profiling Radar (CPR) instruments. This paper provides an overview of the MSI cloud mask algorithm (M-CM) being developed to derive the cloud flag, cloud phase and cloud type products, which are essential parameters for the cloud optical and physical properties (M-COP) as well as for the aerosol optical properties (M-AOT). The MSI cloud mask algorithm has been applied to simulated test data from the EarthCARE end-to-end simulator and satellite data from the Moderate Resolution Imaging Spectroradiometer (MODIS) as well as from the Spinning Enhanced Visible Infra-Red Imager (SEVIRI). Verification of the MSI cloud mask algorithm to the simulated test data and the sophisticated cloud products from SEVIRI and MODIS demonstrates a good performance of the algorithm. Some discrepancies are found however, for the detection of thin cirrus clouds over bright surfaces like desert or snow. This will be improved by tuning of the thresholds once real observations are available."*

From my perspective a description of the M-CM algorithm is provided together with a first validation. The terms "validation" and "verification" should not be used interchangeably as they describe different things. Verification determines if the algorithm developed is suited to the specified requirements. This can be done by testing selected scenes. However, a profound validation should contain a quantitative analysis of a larger and well suited test data set that will follow after the launch as explained in the conclusion. Please check and correct the appropriate usage of these terms throughout the manuscript.

*Thanks, we agree and changed "validation" to "verification" in the paper mostly in chapter 3. The subtitles have been adjusted.*

l.14 Comma is missing after "However". Please check this throughout the manuscript.

*Done.*

l.34 "a a decision tree". Delete "a"

*Done.*

l.122 "makes no sense" Please be more specific. Why does it make no sense? Is a probability more useful for the user's applications as it provides a confidence measure or is there another reason? Please rephrase l.121-124 for clarity.

*We rephrased it to:*

*"The thresholds rely on the assumption that spectral signatures of cloud-free pixels and pixels covered by different cloud types differ. As the thresholds vary globally only the upper (cloudy) and lower (cloud-free) limits of the thresholds are defined and a linear function is used to determine the probability of how close the observation is to the limits. Furthermore, the probability of being cloud-free from the applied tests is combined to an overall probability which may provide, in combination with the number of applied tests a measure of the confidence of the result. From the overall probability a binary cloud mask indicating if a pixel is cloudy or not is derived with four levels of confidence: clear, probably clear, probably cloudy and cloudy."*

l.130 How are these surface dependent thresholds derived?

*The surface dependent thresholds (surface like: ocean, land, desert, snow, ice) are defined before and provided in the configuration data of the M-CLD processor. This allows adaptations of the thresholds at a later stage without changing the processor software.*

l.180 Please rephrase the sentence for clarity. Is the following correct?

*Thanks, we have taken your suggestion:*

*The same applies for the tri-spectral brightness temperature difference test. Further investigation is needed to define the base threshold, which is strongly dependent on surface and water vapor.*

l.190 "probabilities that the pixel is clear" --> probabilities of a pixel being clear

*Done.*

l.237 "the the" --> that the

*Done.*

l. 269 "synthetic atmospheric test scenes created with" atmospheric test scenes created synthetically with …

*Done.*

Fig. 6 Figure subtitle >0.1 disagrees with caption and text (≥ 0.1)

*Thanks, the subtitle of the Fig 6 has been corrected.*

l. 303 "difference" is not true. The term "confusion matrix" would be more appropriate.

*Replaced.*

l. 307 "while the true cloud flag". This may be misleading as the 3D simulations are not ground truth and are highly dependent on the assumptions (l. 276). Might be better to use the word test or simulated cloud flag instead.

*We revised the formulation and found that the term model output is already misleading. As we use the 3D model fields not "the output". The extinction profiles are the 3D model fields, which build the base for the simulation of the synthetic test scenes. Therefore, we changed "output" to "input fields" and "true" to "reference".*

l. 315 "data form the Copernicus" –> data from the Copernicus

*Done.*

l. 351 "Twelve groups" what is meant by this? Where do the 12 algorithms originate from?

*The 12 algorithms are different scientific institutions, like EUMETSAT central facility, the Nowcasting SAF and the Climate Monitoring SAF.*

*We changed the phrase to: "Different scientific institutions (e.g., EUMETSAT central facility, the Nowcasting SAF and the Climate Monitoring SAF) provided cloud mask data for the SEVIRI disk for the intercomparison study."*

---

## Author Comment (AC2)

**Reply to Anonymous Referee #2**

We would like to thank Anonymous Referee #2 for their helpful comments. Below are the original comments in normal with our responses in *italic* text.

**Comments to the author**:
This paper describes the EarthCARE MSI cloud mask (M-CM) algorithms, showing algorithm descriptions and validations. Validations has been performed by using, synthetic dataset, versus MODIS, versus SEVIRI. This paper is well organized and reliable. However, some minor improvements will be needed before accepting this paper. Reviewer recommends that this paper to be accepted with some minor modifications.

(1) Line 27.

Could you please mention about how large of uncertainties can be lead. Please add some reference papers?

The answer to that can be rather complex and difficult. *The cloud masking method used here is based on multispectral sorting of data rather than numerical calculation. There is no direct measure on how uncertain the cloud mask retrieval is. But, to get uncertainties, comparison studies and intercomparison studies have been done like eXercise (Skakun et al., 2022, Zekoll et al. 2021). We also take part in the ICWG intercomparison study (Wu et al., Fig 9), which shows a difference in the cloud fraction up to 30% for the SEVIRI disk. Other studies used comparisons with active instruments. For instance, the ASTER cloud mask for optical thin clouds over ocean gives an uncertainty of 2,7% (Mieslinger et al.). What we meant more is that cloud masking is the fundamental step for the following retrievals like cloud or aerosol properties. As the aerosol properties, which rely on the cloud-free pixel. If thin clouds are not detected by the cloud mask. The aerosol optical thickness will increase.*

*We changed: "Small errors in the cloud mask can lead to large uncertainties, misrepresentations and misinterpretations in subsequently derived products. Many different cloud detection techniques have been developed in the past." to*

*"If for instance cloudy areas are misclassified as clear or vice versa this could negatively impact subsequent retrievals of aerosol or cloud optical properties which underlies the importance of an accurate cloud masking algorithm. "*

(2) Line 77.

What is "a histogram-based scene analysis"? Please clarify this point.

*With the histogram-based scene analysis, the Maximum-Likelihood method (M-Ctype) is meant, since this method uses the three-dimensional histogram (Fig 4) of the observations (VIS, SWIR-2 and TIR-2) to determine the cloud type. We update this sentence: "It exploits the*

*full spectral information content of the MSI instrument, and also includes a histogram-based scene analysis" to "It exploits the full spectral information content of the MSI instrument (e.g., the cloud type is determined using three-dimensional histograms of the VIS, SWIR-2 and TIR-2 channels)."*

(3) From Line 111 to 115.

Please unify the symbol that means "phi".

*Done.*

(4) Line 138.

Please define "the sun glint angle" by schematic figure or formulation.

*The formulation has been given on page 5. We now give a reference to the equation. "Ocean pixels affected by sun glint also apply thresholds based on the 0.865 μm-channel, but the thresholds are calculated depending on the sun glint angle (see Eq. 2)"*

(5) Equation (3) and (4)

Why multiply 100?

*Indeed, the 100 was not very meaningful. Thanks for pointing it out to us, this was an old relict in the Algorithm Theoretical Baseline Document (ATBD) of M-CLD from the first code.*

(6) Line 253, and 254.

Please write formulations of the NDVI and the NDSI.

*Equation has been added.*

(7) Line 299.

That does a sentence "The M-CF algorithm yields a cloud fraction of 50%" mean? Please clarify this point *(see Fig.7)*.

*We averaged the cloudy pixel from the M-CF product for the Halifax scene and get an average of 50% cloud fraction.*

*We changed the sentence to: ": Using M-CF, a cloud fraction of 50 % is determined for this scene (see Fig.7)."*

(8) Line 329.

I'm confusing. My understanding is that, Fig.8 MODIS cloud flag was generated by using MODIS-Level1b data with use of the M-CLD processor as noted Line around 315. So, spectral channels used in both MODIS cloud flag and MSI cloud flag is the same (is it true?). However, this sentence mentioned that "the MODIS cloud tests are based on much more spectral channels". I imagine that two examinations are mixed in this topic. Please explain more.

*In order to make it clearer, the MSI M-CLD processor used only a subset of the 36 MODIS channels. The seven MSI channels are taken from the comparable MODIS channels. The cloud flag from MODIS is the standard L2 MODIS product, which uses also additional MODIS channels.*

*We changed the phrase to: "We have used the MODIS Terra L1b calibrated radiances (MOD021KM) of seven similar channels to MSI and global forecast data from the Copernicus Atmosphere Monitoring Service (CAMS) as input for the M-CLD processor. For verification of our results, however we use the standard MODIS L2 cloud product which makes use of more spectral channels compared to MSI. "*

---

## Author Response (AR2)

**Reply to Editor**

We would like to thank the editor for his helpful comments and time. Below are the original comments in normal with our responses in *italic* text.

**Comments to the author**:

1. Abstract: it would be helpful to clarify that M-COP and M-AOT are downstream EarthCARE products, e.g. replace "which are essential parameters for the cloud optical and physical properties (M-COP) as well as for the aerosol optical properties (M-AOT)" to "which are essential inputs to downstream EarthCARE algorithms providing cloud optical and physical properties (M-COP) and aerosol optical properties (M-AOT)".

*Thanks, done (new line 8).*

2. Reviewer #2 requested changes at the original line 27: your reply provided the new text but the new manuscript does not include this text. Please add it, and also include some of the papers that you reference in the reply.

*Sorry, we include now at the new line 27: "If for instance cloudy areas are misclassified as clear or vice versa this could negatively impact subsequent retrievals of aerosol or cloud optical properties which underlies the importance of an accurate cloud masking algorithm. Different comparison studies and intercomparison studies have been done like the Cloud Masking Intercomparison eXercise (CMIX) to evaluated the cloud masking algorithms (Skakun et al., 2022; Zekoll et al., 2021)."*

3. Reviewer #2 requested changes at the original line 77: your reply provided the new text but again the new manuscript does not include this text. Please add it.

*We apologise for that. We include now at new line 79: "It exploits the full spectral information content of the MSI instrument (e.g., the cloud type is determined using three-dimensional histograms of the VIS, SWIR-2 and TIR-2 channels)."*

4. L118: "way how" -> "way that". Also you probably want to replace "adopted" with "adapted".

*Thanks, done (new line 120).*

5. L120: "As the thresholds varying globally only" -> "As the thresholds vary globally, only".

*Thanks, done (new line 122).*

6. L121: "determine the probability how close the observation is to the limits" is not clear. Perhaps you mean "determine the probability that a cloud is really present based on how close the observation is to the limits"?

*This is true. We changed it (new line 123).*

7. Reviewer #1 asked how the surface-dependent thresholds were defined (L130 in the original manuscript) but your reply does not answer this (you just say they are "defined before") and no

change appears to have been made to the paper. Please explain in the manuscript how these thresholds were defined.

*Sorry, for being not clear enough. The cloud flag algorithms is adapted from the MODIS cloud mask algorithm and we started with MODIS thresholds as a first guess. But the values has been already tuned based on simulated MSI properties and further adaptation will follow in a later stage, with real data. We added (new line 130):*

*"These thresholds are initially taken from the MODIS cloud mask algorithm. These thresholds have been tuned based on simulated MSI properties, while further adaptions are planned at a later stage, when actual MSI data will become available."*

8. L158: "The pixels identified as cloudy if" -> "The pixel is identified as cloudy if". Moreover, the two sentences on lines 155-157 are each their own paragraph. Generally speaking a paragraph should be composed of more than one sentence, so I suggest these two sentences are combined into the paragraph beginning on line 158. Likewise line 166 is another one-sentence paragraph.

*Thanks, done (new line 161).*

9. Eqs. 11 and 12: NDVI and NDSI are acronyms so should be rendered in upright roman in equations, just like in the text. In Latex equations this can be done with \mathrm{NDVI}.

*Thanks, done (new line 250).*

10. L304. "confusion matrix" may not be familiar to some readers, so I suggest this is replaced by "hits, misses, false alarms and correct negatives (i.e. occurrences of elements of the confusion matrix)".

*We added: new Line 298: "Figure 7 illustrates the performance of the M-CF cloud flag compared to the reference cloud flag (using a threshold of COT >=0.1) by showing the results of the confusion matrix (e.g., true positive, true negative, false positive, false negative)."*

*Figure 7: "M-CF cloud flag (left) and confusion matrix (right) indicating the classification performance (e.g. true cloudy, true clear, false cloudy, false clear) of the binary M-CF and the reference cloud flag (using a threshold of COT >= 0.1)."*

11. L316: "(MOD021KM) seven similar" -> "(MOD021KM) of seven similar" (your reply to reviewer #2 included the "of", but not your new text).

*Sorry, done (new line 311)).*

12. Many of your references say "to be submitted" but I think all but one (Eisinger et al.) has now been submitted, so you can replace these with the proper references including the DOI.

*Yes, we updated all. We not sure what we should do with Eisinger et al. paper.*